# CNNs as Inverse Problem Solvers and Double Network Superresolution

## Abstract

In recent years Convolutional Neural Networks (CNN) have been used extensively for Superresolution (SR). In this paper, we use inverse problem and sparse representation solutions to form a mathematical basis for CNN operations. We show how a single neuron is able to provide the optimum solution for inverse problem, given a low resolution image dictionary as an operator. Introducing a new concept called Representation Dictionary Duality, we show that CNN elements (filters) are trained to be representation vectors and then, during reconstruction, used as dictionaries. In the light of theoretical work, we propose a new algorithm which uses two networks with different structures that are separately trained with low and high coherency image patches and show that it performs faster compared to the state-of-the-art algorithms while not sacrificing from performance.

## 1 Introduction

Recent years have witnessed an increased demand for superresolution (SR) algorithms. Increased number of video devices boosted the need for displaying high quality videos online with lower bandwidth. In addition, the social media required the storage of videos and images with lowest possible size for server optimization. Other areas include 4K video displaying from Full HD broadcasts, increasing the output size for systems that have limited sized sensors, such as medical imaging, thermal cameras and surveillance systems.

SR algorithms aim to generate high-resolution (HR) image from single or ensemble of low-resolution (LR) images. The observation model of a real imaging system relating a high resolution image to the low resolution observation frame can be given as

$$g = SHf + n \tag{1}$$

where $\mathbf{H}$ models the blurring effects, $\mathbf{S}$ models the downsampling operation, and $\mathbf{n}$ models the system noise. The solution to this problem seeks a minimal energy of an energy functional comprised of the fidelity of the estimated image $\hat{f}$ to the observational image $f$.

State-of-the art algorithms that are addressing SR problem can be collected under *Dictionary learning based methods* (DLB) and *Deep learning based methods* (DLM) categories. Although SR problem is an inverse problem by nature, performance of other methods such as *Bayesian* and *Example based methods* have been surpassed which is the reason why they are not included in this work. Also the SR problem has never been directly dealt with inverse problem solutions as in Combettes & Wajs (2005) Daubechies et al. (2004) DLB are generally solving optimization problems with sparsity constraints such as Yang et al. (2008) Yang et al. (2010) and $L_2$ norm regularization as in Timofte et al. (2013). The main concern of DLB is creation of a compact dictionary for reconstruction of high resolution (HR) image. Although useful, DLB methods become heavy and slow algorithms as reconstruction performance increases. Recent advances on GPUs have fueled the usage of convolutional neural networks (CNNs) for SR problem. CNN based algorithms such as Dong et al. (2014) and Kim et al. (2016) have used multi-layered networks which have successfully surpassed DLB methods in terms of run speed and performance. State-of-the art algorithms also use Perceptual Loss (PL) to generate new textures from LR images Leibe et al. (2016). By uniting PL and generative networks, photo realistic images can be generated Ledig et al. (2017). PL minimization based algorithms are visually superior to MSE minimization based ones. Stability of such

algorithms have been improved since they have been first proposed Goodfellow (2017). Although, stability issue is not yet completely addressed for generative networks

In Bengio et al. (2013) authors have described representation learning as a manifold learning for which a higher dimensional data is represented compactly in a lower dimensional manifold. They have discussed that the variations in the input space is captured by the representations, for which we are explaining the mechanism at work.

Though CNNs are successful for SR problem experimentally, their mathematical validation is still lacking. We summarized the contributions of this work.

- We show that neurons solve an Iterative Shrinkage Thresholding (IST) equation during training for which the operator is dictionary matrix constructed from LR training data. The solution yields a representation vector as the neuron filters. Contrary to the discussion in literature for which an encoder-decoder structure is needed to obtain and use representations, we claim that the filters themselves become the representations.

- We describe a new concept namely *Representation Dictionary Duality* (RDD) and show that neuron filters act as representation vectors during training phase. Then in the testing phase, filters start acting as dictionaries upon which the HR reconstruction is made layer by layer. This is a concept which helps us analyze CNNs with sparse representation and inverse problem mathematics.

- After analyzing a neuron with inverse problem and DLB solutions and discussing how the entire network operates during training, we propose a new network structure which is able to recover certain details better, faster without sacrificing overall performance.

Rest of the paper organized as follows: in section 2 we refer to related literature for different areas of research. Section 3 ties previous work into our analysis of CNNs. In section 4 we propose a new network for SR problem. In section 5 we give experimentation results.

## 2 RELATED WORK

### 2.1 ANALYTIC APPROACHES

Solution to eq. 1 is inherently ill-conditioned since a multiplicity of solutions exist for any given LR pixel. Thus proper prior regularization for the high resolution image is crucial. The regularization of the inversion is provided with a function, **reg**, which promotes the priori information from the desired output, **reg** takes different forms ranging from $L_0$ norm, Tikhonov regularization to orthogonal decomposition of the estimate. Denoting the SH matrix in eq. 2 by **K**, the regularized solution is given by

$$\hat{f} = \underset{f}{argmin} \frac{1}{2}||Kf - g||_2^2 + reg(f) \tag{2}$$

In Daubechies et al. (2004) authors have used sparsity promoting regularization and Combettes & Wajs (2005) have inspected various proximity mapping functions for solutions of inverse problems with projection onto convex sets. The application of convex analysis results (Combettes & Wajs (2005)) to the linear inverse problem, involves iterations which result in so called Iterative Shrinkage/Thresholding (IST). So, the solution to the inversion of eq. 2 with an $L_1$ norm regularization function can be obtained by the help of Moreau proximity operator as

$$f^n = prox_{b||.||}(f^{n-1} + b.K^T(g - Kf^{n-1})) \tag{3}$$

Where a class of proximity operators are defined, the special function for the case of $L_1$ regularization is soft thresholding function also known as shrinkage operator.

$$prox_{b||.||}f = \left\{ \begin{array}{ll} (1 - \frac{1}{||f||})f & if ||f|| \geq b \\ 0 & otherwise \end{array} \right\} = sign(f).\max(|f| - b, 0) \doteq soft_b(f) \tag{4}$$

Notice that $K^T(g - Kf^{n-1})$ is the negative gradient of data fidelity term in the original formulation. Therefore the solution for the inverse problem using IST iterations is obtained in a gradient descent type method thresholded by Moreau proximity mapping which is also named as Proximal Landweber Iterations. Daubechies et al. (2004) have proposed the usage of non-quadratic regularization constraints that promote sparsity by the help of an orthonormal (or overcomplete) basis $\varphi_l$ of a Hilbert space. For the problem defined in eq. 2 it is proposed to use a functional $\phi_{b,p}$ as

$$\Phi_{b,p}(f) = ||Kf - g||^2 + \sum_{\forall l} b_l|\langle f, \varphi_l \rangle|^p \tag{5}$$

For the case when p = 1, a straightforward variational equation can be obtained in an iterative way.

$$\langle f^n, \varphi_l \rangle = soft_b(\langle f^{n-1}, \varphi_l \rangle + \langle K^T(g - Kf^{n-1}), \varphi_l \rangle) \tag{6}$$

Iterations over the set of basis functions can be carried out in one formula

$$f^n = Z_b(f^{n-1} + K^T(g - Kf^{n-1})) \tag{7}$$

where

$$Z_b(x) \doteq \sum_{l \in \Gamma} (\langle x, \varphi_l \rangle)\varphi_l \tag{8}$$

which can be seen as a method to file the elements of $\mathbf{x}$ in the direction of $\varphi_l$. Daubechies et. al. have proven that the solution obtained by iterating $\mathbf{f}$ is the global minimum of the solution space. The solution will reach to an optimum point if $\mathbf{K}$ is a bounded operator satisfying $||Kf|| \leq C||f||$ for any vector $\mathbf{f}$ and some constant $\mathbf{C}$.

*We will use this result in proving that neurons in a CNN architecture are able to reach to the optimum solution for SR problem by solving for the exact same eq. 7.* A similar work is conducted by Gregor & LeCun (2010). They have proposed a Learned IST algorithm which can be seen as a time unfolded recurrent neural network. Later Bronstein et al. (2012) have discussed that LISTA and their own algorithms that extend LISTA are not mere approximations for an iterative algorithm but themselves are full featured sparse coders.

Our work diverges from theirs in showing how a convolutional neural network is able to learn image representation and reconstruction for SR problem inside network parameters. We will unite inverse problem approaches, DLM and DLB methods in a representation-dictionary duality concept.

## 2.2 Data Driven Approaches

### 2.2.1 Dictionary Learning Based Superresolution

Instead of approaching the superresolution problem to directly invert an observation model, DLB learn mappings from LR to HR training images based on a dictionary. The algorithms jointly solve for a compact dictionary and a representation vector. Sparse representation has been applied to the dictionary learning based SR problem. An LR image is sparsely represented by an LR dictionary. The representation vector is either directly or by some changes applied to an HR library for reconstruction of HR image. DLB algorithms both solve for creating dictionary and solve for a representation vector for any input.

The K-SVD algorithm Aharon et al. (2006) is one of the keystones of dictionary learning for the purpose of sparse representation. Aharon et. al. have proposed the usage of a compact dictionary $\mathbf{D}$, from which a set of atoms (columns or dictionary elements) are to be selected via a vector $\mathbf{f}$ and the combination of these atoms is constrained to be similar to a patch (or image) $\mathbf{g}$ via $||g - Df||_p \leq \varepsilon$. If the dimension of $\mathbf{g}$ is less than that of matrix $\mathbf{D}$ and if $\mathbf{D}$ is full-rank matrix then there are infinitely many solutions to the problem therefore a sparsity constraint is introduced.

$$\min_f ||f||_0 \ s.t. \ ||g - Df||_2 \leq \varepsilon \tag{9}$$

The $L_0$ norm gives the number of entries in $\mathbf{f}$ that are non-zero. The usage of compact dictionaries for SR problem is introduced in Yang et al. (2008). The authors have used the approach of K-SVD.

The optimization of $L_0$ norm regularized equation is hard and a closed form solution might not be available. For the case when **f** is sufficiently sparse, eq. 1 can be approximated by $L_1$ norm. The solution of such an equation can be obtained by Lagrange multipliers.

$$\min_{f} \lambda ||f||_1 + \frac{1}{2}||Df - g||_2^2 \qquad (10)$$

During learning phase the library D is initialized by random gaussian noise and an iterative algorithm between a batch representation matrix Z and dictionary D refines the dictionary while maintaining sparsity for representation vectors of training set. Yang et al. (2008) uses two dictionaries, one for LR representation, one for HR reconstruction as described in the beginning of this chapter.

Timofte et al. (2013) have proposed the usage of $L_2$ norm instead of $L_1$ norm for even faster computations. Although usage of $L_2$ norm eliminated the sparsity constraint from the equation it will play a role in understanding how CNNs work in later chapters.

### 2.2.2 Convolutional Neural Networks

The mapping between the high and low resolution images can also be found by convolutional networks (Dong et al. (2014), Kim et al. (2016)).

The activation function plays an important role in neural network training. In many state-of-the-art algorithms major functions such as tanh and softmax have been replaced by rectified linear units Maas et al. (2013) that are linear approximations of mathematically complex and computationally heavy functions. Glorot et al. (2011) has empirically shown that by using rectified activations the network can learn sparse representations easier. For a given input, only a subset of hidden neurons are activated, leading to better gradient backpropagation for learning and better representations during forward pass. Especially sparse representation has been shown Glorot et al. (2011) to be useful. Sparsity constraint provides information disentagling which allows the representation vectors to be robust against small changes in input data.

Romano et al. (2017) uses gradient information to separate image pixels during interpolation. Separation is done according to three properties namely, strength, coherence and angle. A low strength and coherence signifies as lack of content inside the patch. A high strength but low coherence signifies corner or multi directional edge information. High strength and high coherence signifies a strong edge. Especially the coherence information will play an important role in section 3.

Dong et al. (2014) have provided the earliest relation of CNNs to Sparse Representation. In their view outputs of the first layer constitute a representation vector for a patch around each pixel in LR image, second layer maps LR representations to HR representation vectors and the last layer reconstructs HR image using 5x5 sized filters (or atoms if we have used the jargon of sparse representations). Although this idea qualitatively maps CNNs as a solution method for sparse representation problem, we will now show a more complete understanding with mathematical background. Figure 5 in Appendix shows how SRCNN algorithm works.

## 3 Relation of Inverse Problem, Sparse Representation and CNNs

Even though CNNs yield very good estimates of superesolved images, the connection between inversion of observation model and activation of neurons in CNNs is missing. In this section, we will show the relation between the inverse problem solutions and sparse representation to CNNs. Figure 1 summarize how we are connecting all previous work to CNNs. Considering single neuron we are going to generalize the solution.

For the training phase of CNNs, LR images are fed into the network for forward pass. The resulting image from the network is compared against a ground truth HR image and the error is backpropagated. Since the input image is convolved by the neuron filter, its size should be larger than the size of the output to prevent boundary conditions. This will not be a problem in our case, as stated by Kim et al. (2016), the results from a deep residual network are not spoiled even at the edges of the images. The convolution operation can be carried out in an algebraic manner. Let us assume that we are operating on a patch of LR image, that is named as superpatch. The superpatch is divided into chunks, that are named as subpatches, which have the same support as the filter. Filter and each

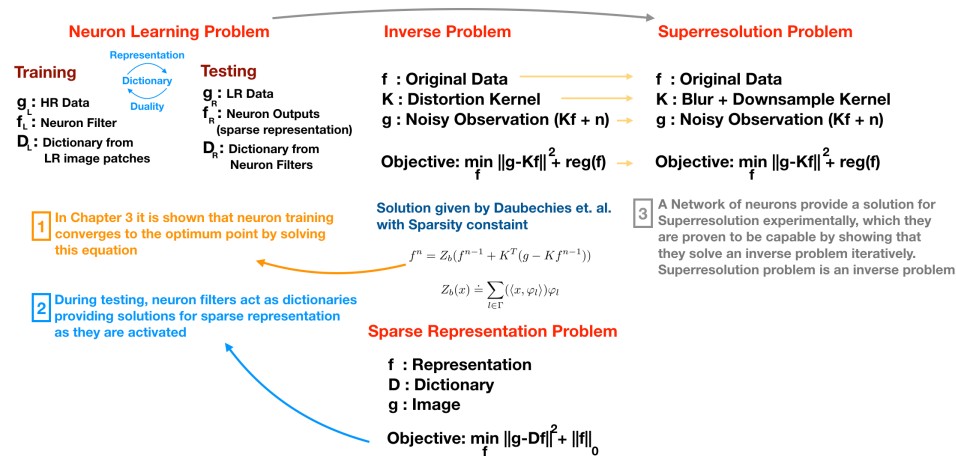

Figure 1: Connecting the Dots

subpatch is vectorized in column stack method. Vectorized subpatches are concatenated to form a matrix. Figure 6 in Appendix visualizes this procedure.

During training CNN solves the mapping of LR input to HR image in training set. The product of the network is going to be a mapping, $\mathbf{f}_L$, from LR superpatch, that is collected under $\mathbf{D}_L$, to HR patch, $\mathbf{g}_L$. Therefore the vector $\mathbf{f}_L$ will be the neuron filter, i.e. the only variable for the training phase, $\mathbf{D}_L$ will be concatenated subpatch matrix, each subpatch in vectorized form will be named as $\varphi_l$. The vector $\mathbf{g}_L$ will be a patch from HR image as in Figure 6. We will show that the CNN operations solve for the same equation as in eq. 7. Since the $\mathbf{D}_L$ matrix satisfies the boundedness constraint on the operator of the equation, the solution will be optimum.

We now modify gradient descent type learning process. Convolution of subpatches with the filter can be algebraically written as $D_L f_L$. Then,

$neuron\ output = Soft_b(D_L f_L)$
$output\ error = g_L - Soft_b(D_L f_L)$
$output\ MSE = ||g_L - Soft_b(D_L f_L)||^2$

Taking the gradient of MSE with respect to $\mathbf{f}_L$ is tricky. When an element of $D_L f_L$ vector lies below the bias, the result will be zero causing the gradient to be zero. We modify the equation by changing the bias vector to $b\prime$ to enable us to use MSE gradient formulation in $Soft_{b\prime}(D_L^T(g_L - D_L f_L))$. This is a valid insertion since the addition of $\mathbf{g}_L$ and multiplication of $\mathrm{D}_L^T$ are linear operators that can be used to scale elements of original bias vector $\mathbf{b}$. Then,

$filter\ update = f_L + Soft_{b\prime}(D_L^T(g_L - D_L f_L))$
$(change\ bias) = Soft_{b\prime\prime}(f_L + D_L^T(g_L - D_L f_L))$
$(introduce\ \varphi_l) = Soft_{b\prime\prime\prime}\langle(f_L + D_L^T(g_L - D_L f_L)), \varphi_l\rangle$

Then the summation of thresholded decompositions is given as $Z_b(f_L + D_L^T(g_L - D_L f_L))$ which is the same as eq. 7 where $Z_b$ is defined in eq. 8. Therefore gradient descent type learning of a single neuron is guarantied to reach an optimal solution using eq. 3.

For the testing phase, a new representation - dictionary duality (RDD) concept is proposed. RDD concept states that the representation vectors learned during the training phase can be used as atoms of a dictionary for the testing phase. The cost function that is minimized by CNN training (learning) yields a representation vector as the neuron filter, for which the dictionary is matrix $\mathrm{D}_L$ and the target is HR image patch. *During testing (scoring, reconstruction) phase, resulting representation vectors (filters) from a layer of neurons turn into a dictionary (later named as $D_R$) upon which the*

*reconstruction of HR image is carried out.* A similar idea is proposed by Papyan et al. (2016) and Papyan et al. (2017) stating that each layer output which is a representation for inputs of previous layer, can also be seen as an input to be represented by the next layer. The authors have argued that each layer output will contain a structure which can be represented by a convolutional sparse coding (CDC) layer. A CDC layer is essentially a CNN layer. The difference of our RDD is that we use the idea that dictionaries and representations swap roles during training and testing (forward pass). Also during training, inputs to each layer is perceived as a dictionary for the next layer, contrary to previously proposed perception of Papyan et al. (2016). Following the idea of RDD, the neuron filter, previously named as $\mathbf{f}_L$, can be viewed as an atom of a dictionary consisting of many other neuron filters. During testing period, the filters are vectorized and concatenated to form the dictionary matrix $\mathbf{D}_R$, the vector $\mathbf{g}_R$ will be the input image this time and the $\mathbf{f}_R$ vector will be the neuron outputs, which will be the representation vector of input image in terms of the dictionary atoms, i.e. the neuron filters. The mathematical insight for this is again given in eq. 3. Considering the initial condition for the equation, during testing phase, $f_R^0$ can be assumed as zero and the $f_R^1$ is going to be the representation vector provided with

$$
\begin{aligned}
f_R^1 = prox_{b||.||}(f_R^0 + b.D_R f_R^0||) &= prox_{b||.||}(D_R^T g_R) \\
&= soft_b(D_R^T g_R) = \max(D_R^T g_R - b, 0)
\end{aligned}
\tag{11}
$$

Again we reach to the conclusion that the ReLU operators provide the representation vector, $\mathbf{f}_R$, for the input image, $\mathbf{g}_R$, given the trained filter values collected under $\mathbf{D}_R$. We have demonstrated this feature in experimentation chapter. To provide a visually meaningful example we have used a training set that contain highly coherent edges with a narrow orientation range. RDD is visually apparent only for the first layer and for training sets with similar information content. Deeper layers feed from previous layer's outputs therefore it is hard to demonstrate for all layers. Also while training with a more general training set, an observer will not see any patterns in learned filters. Seeing apparent features would mean memorization which is a degrading property for a neural network.

To extend the understanding of single neuron to the entire network Theorem 1 will be used from Papyan et al. (2016).

**Theorem 1** Suppose $\mathbf{g} = \mathbf{y} + \mathbf{n}$ where $\mathbf{n}$ is noise whose the power of noise is bounded by $\varepsilon_0$ and $\mathbf{y}$ is a noiseless signal. Considering a convolutional sparse coding structure where $D_l$ is the dictionary for $l^{th}$ layer
$y = D_1 f_1$
$f_1 = D_2 f_2$
.
.
.
$f_{N-1} = D_N f_N$
Let $\hat{f}_i$ be a set of solutions obtained by running a convolutional neural network, or layered soft thresholding algorithm with biases $b_i$ as $\hat{f}_i = soft_i\{D_i^T \hat{f}_{i-1}\} where \hat{f}_0 = g$. Denote $|fmax|$ and $|fmin|$ as absolute maximum and minimum entries of representation vectors. Then assuming for $\forall 1 \le i \le N$
$||f_i||_0 < \frac{1}{2}(1 + \frac{1}{\mu(D_i)}) - \frac{1}{\mu(D_i)}\frac{\varepsilon_{i-1}}{|fmax|}$
where $\mu(D_i)$ is the mutual coherence of the dictionary then

1. The support of the solution $\hat{f}_i$ is equal to the support of $f_i$

2. $||f_i - \hat{f}_i||_2 \le \varepsilon_i$
$where \ \varepsilon_i = \sqrt{||f_i||_0}(\varepsilon_{i-1} + \mu(D_i))(||f_i||_0 - 1)|fmax| + b_i$

The theorem shows that a network consisting of layered neurons could yield the same result as a layered sparse coding algorithm. Therefore a network of neurons, whose optimality for inverse problem solutions has been proven individually, is now proven to reach to a solution for sparse coding. Let us now recall the Landweber equation applied for CNN $f_L = (D_L^T D_L + \mu I)^{-1} D_L^T g_L$ In order to be able to use insights from this equation assume that all neurons in the network are activated for the inputs. For that un-realistic case, the network filters can be convolved among themselves to produce an end point filter, $\mathbf{f}_L$. This is feasible because when all neurons are activated,

their linear unit outputs are going to be the convolution results minus a bias that can be added up at the end, simply enabling the convolution of all filters to be applied in a single instant. A similar work is done by Mallat (2016) to analyze linearization, projection and separability properties of sparse representations for deep neural networks.

The vector $\mathbf{f}_L$ is going to be a normalized projection of $\mathbf{g}_L$ onto LR image domain. Considering the rows of $D_L^T$ matrix, each row is a vectorized subpatch, thus each multiplication result from $D_L^T \mathbf{g}$ is going to be $\langle subpatch, g_L \rangle$ meaning the projection of HR patch onto an LR subpatch. $D_L^T D_L$ matrix have elements of inner products of subpatches such as $\langle subpatch_i, subpatch_j \rangle$. The diagonals of $D_L^T D_L$ matrix, therefore, are normed square of each subpatch. The inverted matrix is going to be mostly composed of diagonals that are inverted normed square values of subpatches. This means that the entire equation calculates the projection of HR patch, $\mathbf{g}_L$, onto the LR image domain. In other words, the result, $\mathbf{f}_L$, consists of scores which measure how similar $\mathbf{g}_L$ vector is to each subpatch from the entire superpatch. If the HR image has content that cannot be recovered by using certain region of LR image, the reconstructed image is going to be inferior. This is due to the violation of overcompleteness assumption. Selection of a larger area for the reconstruction of certain HR patches proves useful because of increased information included into the system that brings the subpatches, or bases, closer to being overcomplete.

This insight provides a method for determining how deep a network should be for certain features. For example when the superpatch and corresponding HR region contains only texture, which can be modeled as gaussian noise, the $\mathbf{D}_L$ matrix becomes linearly independent, meaning easily invertible. Consequently when the training set consists solely of textured images, shallow networks should do as good as deep networks. Then for the testing phase, same filters are used to construct the $\mathbf{D}_L$ matrix and the result of the network is obtained by the same equation without normalization (without the inverse term) this time (since it is already normalized) as in eq. 11, i.e. projecting LR image onto filters' domain. Notice that the error is generally not completely orthogonal to the LR images because of iterative nature of equations. Therefore this is not going to be a meaningless operation. The representations that are learned during training can only be called complete if the data can be completely recovered Ranzato et al. (2006). Since the method by which the network recovers HR details is through inner products, the assertion of RDD seems complete.

In general the training set contains various features with different variances. Therefore the generalization of the new concepts that are introduced here are difficult. Training with different structures enables the constant evolution of neuron filters during training. However to have an activating branch for each feature either the network should have increased number of filters or the network will not converge which can be explained by the manifold hypothesis, as representations not covering the high dimensional input space Bengio et al. (2013).

This is the point where we tie theoretical work into a practical network.

## 4 PROPOSED NETWORK

The discussions from section 3 revealed that using a single training set for a single network is complicating the training process. Because we expect the neuron filters to learn predominant patterns and information from the training set, training a single network either leads to a heavy network with lots of memory requirement or leads to insufficiently learned filters. We are proposing usage of a double network SR (DNSR) for two different data. The data separation is done according to gradient information, dividing data set into low and high coherence sets. For low coherence data which is mainly texture, we have trained a shallower network as in Figure 8. We have used network depth of 10 layers, as tests with shallower or deeper networks slightly turned out to be in favor of 10 layers. High coherence data contains edge and corner information. We trained a deeper network of 20 layers to reconstruct edge information. This is, to the best of our knowledge, the first time proposition of separation of neural network for the purpose of recovering different contents for SR.

In order to satisfy the assumption made in Theorem 1, which concerns the coherence of dictionary elements, we have used skip connections between layers to correlate the outcomes. This is only done in low coherence network due to inherent lack of correlation of dictionary elements which are input LR images, as our RDD explains. Since output of each layer of neurons is an input to next layer, acting as a new dictionary, skip layers qualitatively provide the required increase

in coherence. Usage of skip layers have also been proposed by Mao et al. (2016). The authors have used skip layers in a very deep network (30 layers) to prevent gradient vanishing problem and propagating information between two different structures (conv and de-conv layers). In this work we are using skip layers in a network which is required to be as shallow as practically possible to increase coherence of layer outputs. We also used cross entropy loss besides the MSE loss for low coherence network similar to GAN based algorithms Goodfellow (2017).

We have used bicubically upsampled inputs to the network which is the only pre-processing before neural networks. The aggregation of two separate network are done in a post-processing block because the training operation uses separate validation data for error gradient calculations. We have backprojected the results to upsampled input images and then simply added two outputs by giving more weight to high coherence network.

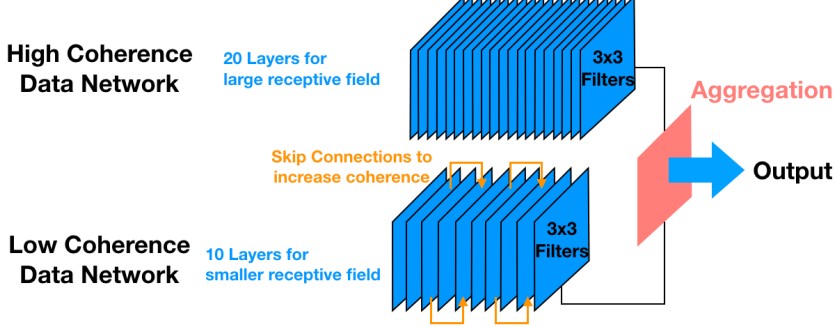

Figure 2: DNSR Structure

# 5 EXPERIMENTAL RESULTS

## 5.1 EXPERIMENTAL SETUP

We have used an Intel i7-4770K CPU Nvidia GeForce GTX 760 GPU computer to run the training and testing operations. Since we do not readily have a GPU implementation, we have given individual run times for each image while comparing speed. Training is completed in 16 hours for high coherence network and 8 hours for low coherence network which is significantly less then the requirement of state of the art algorithms. The run times are as fast as twice the speed of reference model Kim et al. (2016) as reported in Table 1

We have used the same 291 image training set from Kim et al. (2016). In similar fashion we have rotated and scaled the images to create an augmented set. Then we have separated patches into two subsets according to their coherence values obtained from upsampled LR images. Our tests are carried out in scaling factor of 3x.

## 5.2 REPRESENTATION-DICTIONARY DUALITY

We have conducted experiments to test out the RDD proposition which states that the learned filters for neurons resemble to the highlighted features from training data. We have created two separate training set which contained high coherence data with edges of orientation 0-20 degrees and 40-60 degrees. The results were showing that the learned filters for the first layer resemble the predominant features of the training set as in Figure 3 and Figure 4

## 5.3 SEPARATE NETWORKS

The advantage of separate network training is to be able to recover details that otherwise might be dropped out during training due to stronger data. Barbara image details can be clearly seen in Figure 7 in appendix, low coherence network output. The PSNR and SSIM value of Barbara image shows the validity of this example.

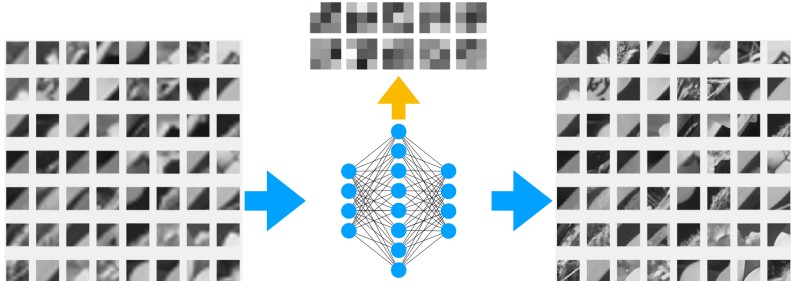

Figure 3: Trained network filters 40-60 degrees oriented data set

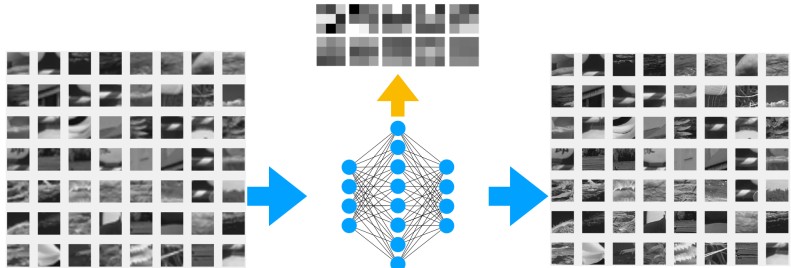

Figure 4: Trained network filters 0-20 degrees oriented data set

Not only coherence but also strength and angle information can be used to divide the training data. Initial experiments with increased number of parallel networks showed that networks with low strength data, which are almost flat patches, did not converge to a useful state after training. Also aggregation of networks trained with different orientation edges became cumbersome and such a network was not feasible for a real time application. We have decided on using two networks whose data are divided according to coherence which was the only option that we could theoretically support.

## 5.4 RESULTS

Numerical comparisons are done only with the referenced algorithm Kim et al. (2016) in Table 1. A comparison with previous DLB and DLM can be found in the reference paper Kim et al. (2016).

The proposed network is faster compared to Kim et al. (2016) due to lighter networks. Since we have split the training set depending on its information content (i.e. textures and edges) both networks require less number of elements to represent the data which yields a faster algorithm.

## 6 CONCLUSIONS

We have proven that a neuron is able to solve an inverse problem optimally. By introducing RDD we have shown that CNN layers act as sparse representation solvers. We have proposed a method that addresses the texture recovery better. Experiments have shown that RDD is valid and proposed network recovers some texture components better and faster than state of the art algorithms while not sacrificing performance and speed. In the future we plan to investigate a content-aware aggregation method which might perform better than simple averaging. We will investigate ways of jointly training or optimizing two networks and including aggregation step inside a unified network. In parallel we are investigating a better network structure for texture recovery. Also we are going to incorporate the initial upsampling step into the network by allowing the network to learn its own interpolation kernels.

|  | Picture | Bicubic | VDSR | | | DNSR | | |
|---|---|---|---|---|---|---|---|---|
|  |  | PSNR | PSNR | SSIM | Time | PSNR | SSIM | Time |
| SET5 | butterfly_GT | 24,0389 | 29,9545 | 0,9677 | 1,1228 | 29,4704 | 0,9640 | 0,6283 |
|  | baby_GT | 33,9302 | 35,3951 | 0,9443 | 4,1837 | 35,3147 | 0,9442 | 2,1753 |
|  | bird_GT | 32,5886 | 36,6726 | 0,9808 | 1,2706 | 36,2698 | 0,9801 | 0,6950 |
|  | head_GT | 32,9069 | 33,9782 | 0,8635 | 1,3000 | 33,9211 | 0,8593 | 0,6689 |
|  | woman_GT | 28,5669 | 32,3509 | 0,9706 | 1,3159 | 32,0630 | 0,9693 | 0,6583 |
| AVERAGE | | 30,4063 | 33,6703 | 0,9454 | 1,8386 | 33,4078 | 0,9434 | 0,9652 |
| SET14 | lenna | 31,6895 | 33,9781 | 0,9796 | 4,1932 | 34,0052 | 0,9796 | 2,3511 |
|  | baboon | 23,2105 | 23,7817 | 0,7243 | 3,9271 | 23,7218 | 0,7187 | 2,0124 |
|  | barbara | 26,2524 | 26,2078 | 0,8039 | 6,6983 | 26,5488 | 0,8073 | 3,3991 |
|  | bridge | 24,4029 | 25,3857 | 0,6417 | 4,2552 | 25,3529 | 0,6389 | 2,0495 |
|  | coastguard | 26,5483 | 27,3816 | 0,5639 | 1,6972 | 27,3916 | 0,5621 | 0,8040 |
|  | comic | 23,1151 | 25,1082 | 0,8356 | 1,4625 | 24,9035 | 0,8292 | 0,7785 |
|  | face | 32,8458 | 33,9690 | 0,8657 | 1,2492 | 33,8832 | 0,8612 | 0,6369 |
|  | flowers | 27,2320 | 30,0112 | 0,9031 | 2,8359 | 29,7569 | 0,8999 | 1,4440 |
|  | foreman | 31,1483 | 34,9947 | 0,9506 | 1,6767 | 34,7941 | 0,9474 | 0,8091 |
|  | man | 27,0104 | 28,7860 | 0,6414 | 4,1589 | 28,6320 | 0,6364 | 2,0660 |
|  | monarch | 29,4332 | 34,7076 | 0,9884 | 6,2657 | 34,0863 | 0,9870 | 3,0428 |
|  | zebra | 26,6298 | 29,4919 | 0,9540 | 3,7495 | 29,5281 | 0,9532 | 1,8033 |
|  | pepper | 32,3859 | 35,2789 | 0,9781 | 4,1047 | 35,1339 | 0,9770 | 2,0480 |
|  | ppt3 | 23,6469 | 27,6872 | 0,8293 | 4,9982 | 27,2313 | 0,7904 | 2,7540 |
| AVERAGE | | 27,5394 | 29,7693 | 0,8328 | 3,6623 | 29,6407 | 0,8277 | 1,8570 |

Table 1: Results

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

# A VISUAL RESULTS

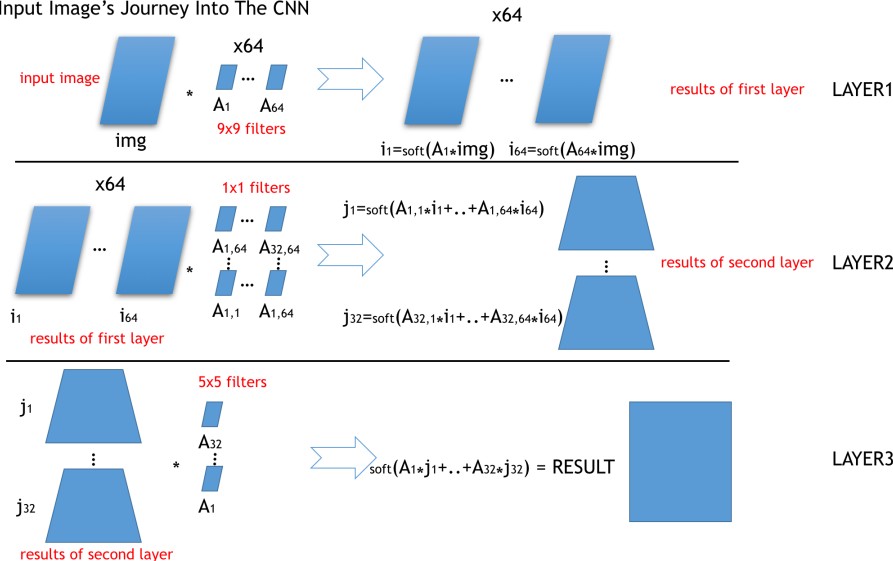

Figure 5: SRCNN Structure

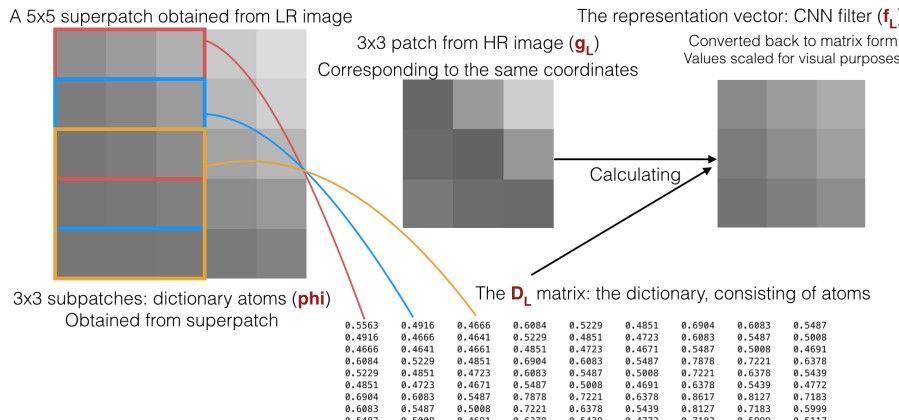

Figure 6: Visualizing Neuron Filter Training Procedure

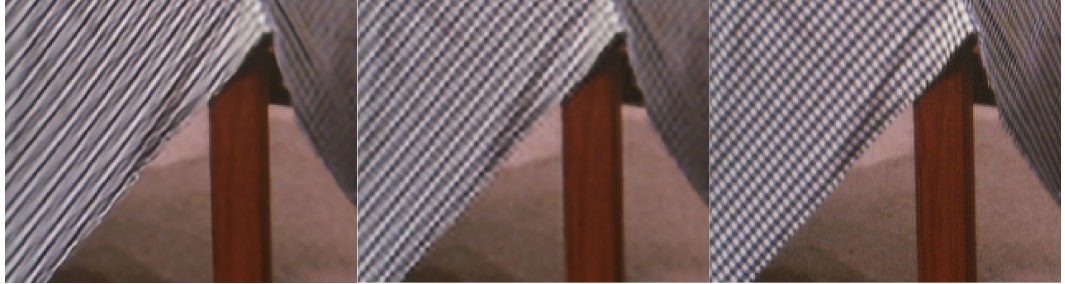

Figure 7: Recovery of texture from Barbara image VDSR(left) DNSR(middle) GT(right)

Detailed visual results are given for different network outputs.

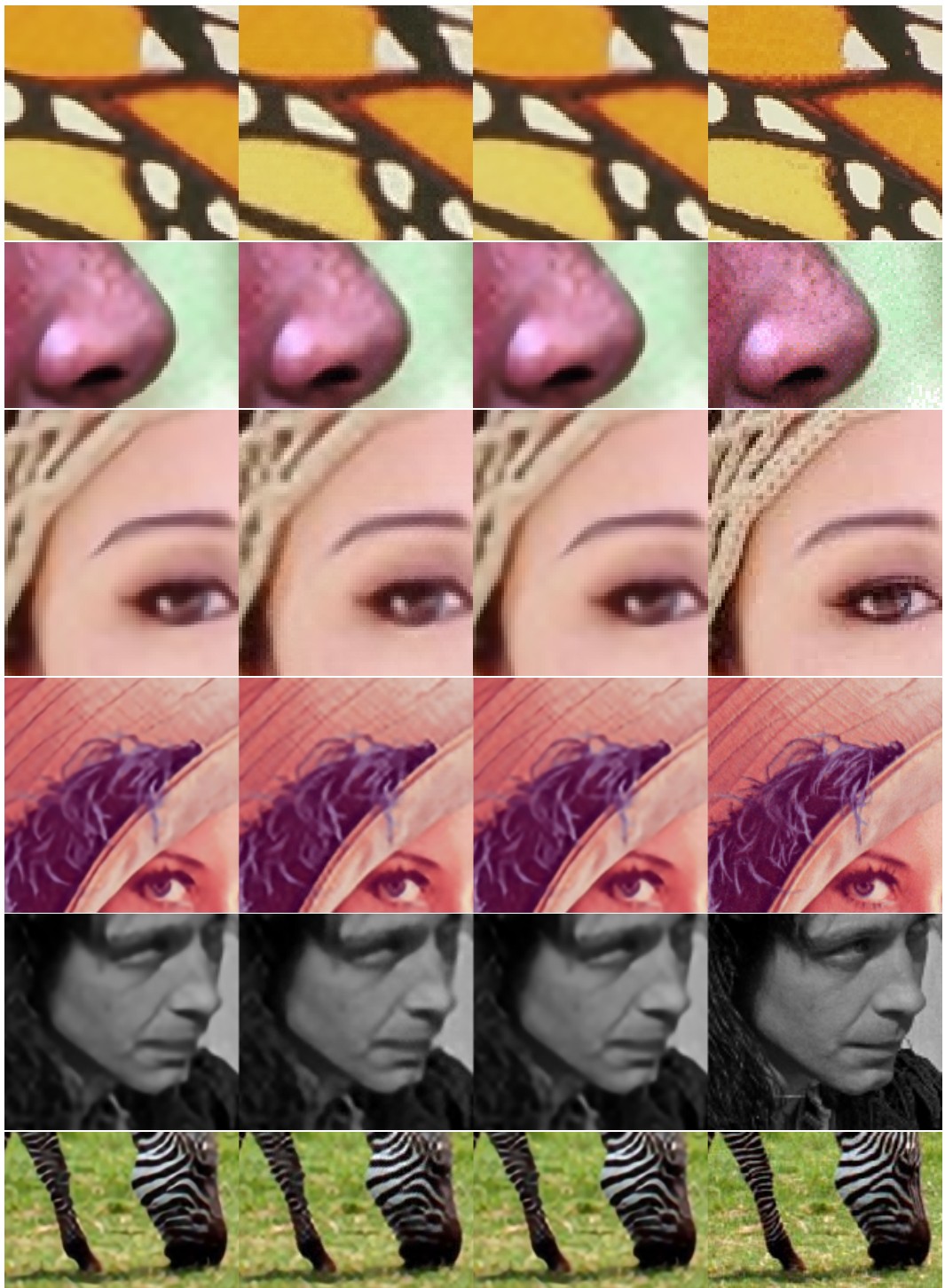

Figure 8: Comparison of different network outputs. From left to right: High and Low Coherence, Aggregation Result and Ground Truth

