# OpenReview forum: "CNNs as Inverse Problem Solvers and Double Network Superresolution"
_ICLR.cc/2018/Conference — Reject_

### Official Review · AnonReviewer2 · 2017-11-25
**Interesting paper bringing up different domains. It could be written more reader friendly.**

**Rating:** 6
**Confidence:** 2

**Review:**

The paper proposes an understanding of the relation between inverse problems, CNNs and sparse representations. Using the ground work for each proposes a new competitive super resolution technique using CNNs. Overall I liked authors' endeavors bringing together different fields of research addressing similar issues. However, I have significant concerns regarding how the paper is written and final section of the proposed algorithm/experiments etc.

Introduction/literature review-> I think paper significantly lacks literature review and locating itself where the proposed approach at the end stands in the given recent SR literature (particularly deep learning based methods) --similarities to other techniques, differences from other techniques etc. There have been several different ways of using CNNs for super resolution, how does this paper’s architecture differs from those? Recent GAN based methods are very promising and how does the proposed technique compares to them?

Notation/readability -> I do respect the author’s mentioning different research field’s notations and understand the complication of building a single framework. However I still think that notations could be a lot more simplified—to make them look in the same page. It is very confusing for readers even if you know the mentioned sub-fields and their notations. Figure 1 was very useful to alleviate this problem. More visuals like figure 1 could be used for this problem. For example different network architecture figures (training/testing for CNNs) could be used to explain in a compact way instead of plain text.

Section 3-> I liked the way authors try to use the more generalized Daubechies et. al. However I do not understand lots of pieces still. For example using the low resolution image patches as a basis—more below. In the original solution Daubechies et. al. maps data to the orthonormal Hilbert space, but authors map to the D (formed by LR patches). How does this affect the provability?

Representation-dictionary duality concept -> I think this is a very fundamental piece for the paper and don’t understand why it is in the appendix. Using images as D in training and using filters as D in scoring/testing, is very unintuitive to me. Even after reading second time. This requires better discussion and examples. Comparison/discussion to other CNN/deep learning usage for super-resolution methods is required exactly right here.

Final proposed algorithm -> Splitting the data for high and low coherence makes sense however coherence is a continues variable. Why to keep the quantization at binary? Why not 4,8 or more? Could this be modeled in the network?

Results -> I understand the numerical results and comparisons to the Kim et. Al—and don’t mind at all if they are on-par or slightly better or worse. However in super-resolution paper I do expect a lot more visual comparisons. There has been only Figure 5. Authors could use appendix for this purpose. Also I would love to understand why the proposed solution is significantly faster. This is particularly critical in super-resolution as to apply the algorithms to videos and reconstruction time is vital.

---

> ### Author Response · Authors · 2017-12-21
> **Answers and Corrections**
>
> Thank you very much for the detailed review of the manuscript.
>
> We have revisited the manuscript to reflect all the reviewers’ comments. The proposed Representation Dictionary Duality concept is explained in detail and the notation inconsistencies throughout the text are corrected. In addition the current literature is updated and the differences between the proposed understanding and the literature is made clear.
>
> Introduction/literature review-> We have referred to the Generative Networks in the revised manuscript.
>
> Notation/readability -> We fixed the notations together with few typos. We added a figure detailing the referenced CNN algorithm SRCNN (Dong et. al.). Also a figure for CNN training procedure is added.
>
> Section 3-> In Daubechies et. al. from our references, the nature of matrix K is defined as a bounded operator between two hilbert spaces. Boundedness is defined according to the formula: for any given vector f from a Hilbert space, if the inequality ||Kf|| \leq C||f|| is satisfied, where C is a constant, then the operator is bounded. The iterative shrinkage algorithm we have referenced from Daubechies et. al. have addressed this issue directly, for cases when the null space of K for a vector f is not zero and its inversion is ill-posed or even ill-conditioned. We have shown that a neuron filter solves the same equation during training and since a library D also satisfies boundedness assumption we know that it will reach to the optimum solution. We now made this clearer in the text.
>
> Representation-dictionary duality concept -> We have moved the appendix A into the text. We assert that, CNN operates as a layered DLB during training and during testing. We have shown that the mechanism by which the CNN learns is through solving an inverse problem. The inverse problem constitutes a bounded operator, matrix D, which is composed of LR patches. Even though the matrix D is different in structure from conventional inverse problem operators, it satisfies the constraints to be used as an operator. The cost function that is minimized by CNN training yields a representation vector as the neuron filter, for which the dictionary is matrix D and the target is HR image patch. Neuron parameters (filters) being the representation vectors instead of an output from a network is a new understanding in the literature. Resulting representation vectors (filters) from a layer of neuron filters turn into a dictionary upon which the reconstruction of HR image is carried out during testing (scoring) phase. This is the core understanding of RDD concept. Using RDD we are able to demystify how a CNN is able to learn and apply reconstruction of HR images for SR problem.
>
> Final proposed algorithm -> We have used strength, coherence and angle information to divide data into 38 networks initially. We have discovered that networks that are trained with low strength data (which are almost flat patches) won’t converge to a meaningful state. We couldn’t handle the separation of angle information while aggregating all the results. Also this was not a feasible network structure to be implemented for a real time, possible video application. So we reduced to using two networks with low and high coherence. The reviewer is absolutely right in asking why 4 or 8 networks have not been used. This was simply due to lack of time. We will strongly consider doing an analysis on this in near future.
>
> Results -> We ran out of space so we had to get rid of all redundant information. We have now added a page of comparison in the appendices. The proposed solution is faster because splitting the data enabled us to train lighter networks, even though one of the networks is as long as the original reference paper (20 layers). We have touched on the subject briefly on chapter 3. We have now added more discussion as to why the proposed solution is faster. And the sole reason we are trying to speed up the algorithm is because we have real time video superresolution application in our future plans. We have not mentioned this in the text plainly because we have not done anything to address multiframe SR yet.

---

### Official Review · AnonReviewer1 · 2017-11-27
**Review of: CNNs as Inverse Problem Solvers and Double Network Superresolution**

**Rating:** 3
**Confidence:** 5

**Review:**

This paper discusses using neural networks for super-resolution. The positive aspects of this work is that the use of two neural networks in tandem for this task may be interesting, and the authors attempt to discuss the network's behavior by drawing relations to successful sparsity-based super-resolution. Unfortunately I cannot see any novelty in the relationship the authors draw to LASSO style super-resolution and dictionary learning beyond what is already in the literature (see references below), including in one reference that the authors cite. In addition, there are a number of sloppy mistakes (e.g. Equation 10 as a clear copy-paste error) in the manuscript. Given that much of the main result seems to already be known, I feel that this work is not novel enough at this time.

Some other minor points for the authors to consider for future iterations of this work:

- The authors mention the computational burden of solving L1-regularized optimizations. A lat of work has been done to create fast, efficient solvers in many settings (e.g. homotopy, message passing etc.). Are these methods still insufficient in some applications? If so, which applications of interest are the authors considering?

- In figure 1, it seems that under "superresolution problem": 'f' should be 'High res data' and 'g' should be 'Low res data' instead of what is there. I'm also not sure how this figure adds to the information already in the text.

- In the results, the authors mention how some network features represented by certain neurons resemble the training data. This seems like over-training and not a good quality for generalization. The authors should clarify if, and why, this might be a good thing for their application.

- Overall a heavy editing pass is needed to fix a number of typos throughout.

References:

[1] K. Gregor and Y. LeCun , “Learning fast approximations of sparse coding,” in Proc. Int. Conf. Mach. Learn., 2010, pp. 399–406.
[2] P. Sprechmann, P. Bronstein, and G. Sapiro, “Learning efficient structured sparse models,” in Proc. Int. Conf. Mach. Learn., 2012, pp. 615–622.
[3] M. Borgerding, P. Schniter, and S. Rangan, ``AMP-Inspired Deep Networks for Sparse Linear Inverse Problems [pdf] [arxiv]," IEEE Transactions on Signal Processing, vol. 65, no. 16, pp. 4293-4308, Aug. 2017.
[4] V. Papyan*, Y. Romano* and M. Elad, Convolutional Neural Networks Analyzed via Convolutional Sparse Coding, accepted to Journal of Machine Learning Research, 2016.

---

> ### Author Response · Authors · 2017-12-21
> **Answers and Corrections**
>
> Thank you very much for the detailed review of the manuscript.
>
> We have revisited the manuscript to reflect all the reviewers’ comments. The proposed Representation Dictionary Duality concept is explained in detail and the notation inconsistencies throughout the text are corrected. In addition the current literature is updated and the differences between the proposed understanding and the literature is made clear.
>
> -We have highlighted the differences of our understanding from that of Papyan et. al. We have not included more foundational papers including Gregor et. al. inside the text plainly to simplify the text. That was a clear mistake and we have now included references into the revised paper. To discuss the differences of our work from what is already published, we highlight few points:
> --Gregor et. al. have used ISTA algorithm and they have successfully implemented iterative algorithm with a time unfolded recursive neural network, which can be seen as a feed-forward network. Then the architecture is fine-tuned with experimental results
> --Bronstein et. al. have worked on a shift of understanding in that, what they present with a neural network is not a regressor that is approximating an iterative algorithm, but itself is a full featured sparse coder.
> -Our work diverges from theirs in showing how a convolutional neural network is able to learn image representation and reconstruction for SR problem. We have united inverse problem approaches, Deep Learning Based and Dictionary Based methods in a representation-dictionary duality concept. We have showed that during training, neuron filters learn from input images as if the input patches constituted a dictionary for representation. Therefore different from literature the neuron parameters (filters) become representations themselves.  And we show that during testing (scoring) learned filters become the dictionaries for reconstruction. This is now made clearer in the text.
>
> -L1 norm minimization is not the crucial part of our work since only capture the mathematical background and optimality of the solutions. We were only repeating how L2 norm minimization based algorithms have defended their reasoning from changing from L1 norm to L2 norm. We edited this part.
>
> -Figure 1 is not wrong, but previous notation changes could have confused the reviewers and we fixed this in revised paper. The f is high res data that is blurred and downsampled with K. The g is the observation therefore we are trying to estimate highres data by estimating f. This figure is used to sum up the different parts that we have brought together. We hoped it would be useful in understanding the crux of the paper.
>
> -Describing the results as “resembling the training data” was an unfortunate choice of words. The purpose of the experiment was to visualize RDD concept which really states that the Network learns predominant features from the training set, not the images themselves. Since we have reduced the training set to a narrow orientation single edged image database, first layer filters tend to be oriented in the same direction which is a visualization of RDD. This does not correspond to resemblance of filters to the data set itself. We have corrected this in the text.
>
> -We corrected the typos.

---

### Official Review · AnonReviewer3 · 2017-11-28

**Rating:** 4
**Confidence:** 4

**Review:**

The method proposes a new architecture for solving image super-resolution task. They provide an analysis that connects aims to establish a connection between how CNNs for solving super resolution and solving sparse regularized inverse problems.

The writing of the paper needs improvement. I was not able to understand the proposed connection, as notation is inconsistent and it is difficult to figure out what the authors are stating. I am willing to reconsider my evaluation if the authors provide clarifications.

The paper does not refer to recent advances in the problem, which are (as far as I know), the state of the art in the problem in terms of quality of the solutions. This references should be added and the authors should put their work into context.

1) Arguably, the state of the art in super resolution are techniques that go beyond L2 fitting. Specifically, methods using perceptual losses such as:

Johnson, J. et al "Perceptual losses for real-time style transfer and super-resolution." European Conference on Computer Vision. Springer International Publishing, 2016.

Ledig, Christian, et al. "Photo-realistic single image super-resolution using a generative adversarial network." arXiv preprint arXiv:1609.04802 (2016).

PSNR is known to not be directly related to image quality, as it favors blurred solutions. This should be discussed.

2) The overall notation of the paper should be improved. For instance, in (1), g represents the observation (the LR image), whereas later in the text, g is the HR image.

3) The description of Section 2.1 is quite confusing in my view. In equation (1), y is the signal to be recovered and K is just the downsampling plus blurring. So assuming an L1 regularization in this equation assumes that the signal itself is sparse. Equation (2) changes notation referring y as f.

4) Equation (2) seems wrong. The term multiplying K^T is not the norm (should be parenthesis).

5) The first statement of Section 2.2. seems wrong. DL methods do state the super resolution problem as an inverse problem. Instead of using a pre-defined basis function they learn an over-complete dictionary from the data, assuming that natural images can be sparsely represented. Also, this section does not explain how DL is used for super resolution. The cited work by Yang et al learns a two coupled dictionaries (one for LR and HL), such that for a given patch, the same sparse coefficients can reconstruct both HR and LR patches. The authors just state the sparse coding problem.

6) Equation (10) should not contain the \leq \epsilon.

7) In the second paragraph of Section 3, the authors mention that the LR image has to be larger than the HR image to prevent border effects. This makes sense. However, with the size of the network (20 layers), the change in size seems to be quite large. Could you please provide the sizes? When measuring PSNR, is this taken into account?

8) It would be very helpful to include an image explaining the procedure described in the second paragraph of Section 3.

9) I find the description in Section 3 quite confusing. The authors relate the training of a single filter (or neuron) to equation (7), but they define D, that is not used in all of Section 2.1. And K does not show in any of the analysis given in the last paragraph of page 4. However, D and K seem two different things (it is not just one for the other), see bellow.

10) I cannot understand the derivation that the authors do in the last paragraph of page 4 (and beginning of page 5). What is phi_l here? K in equation (7) seems to match to D here, but D here is a collection of patches and in (7) is a blurring and downsampling operator. I cannot review this section. I will wait for the author's response clarifications.

11) The authors describe a change in roles between the representations and atoms in the training and testing phase respectively. I do not understand this. If I understand correctly, the final algorithm, the authors train a CNN mapping LR to HR images. The network is used in the same way at training and testing.

12) It would be useful to provide more details about the training of the network. Please describe the training set used by Kim et al. Are the two networks trained independently? One could think of fine-tuning them jointly (including the aggregation).

13) The authors show the advantage of separating networks on a single image, Barbara. It would be good to quantify this better (maybe in terms of PSNR?). This observation might be true only because the training loss, say than the works cited above. Please comment on this.

14) In figures 3 and 4, the learned filters are those on the top (above the yellow arrow). It is not obvious to me that the reflect the predominant structure in the data. (maybe due to the low resolution).

15) This work is related to (though clearly different)  that of LISTA (Learned ISTA) type of networks, proposed in:

Gregor, K., & LeCun, Y. (2010). Learning fast approximations of sparse coding. In Proceedings of the 27th International Conference on Machine Learning (ICML)

Which connect the network architecture with the optimization algorithm used for solving the sparse coding problem. Follow up works have used these ideas for solving inverse problems as well.

---

> ### Author Response · Authors · 2017-12-21
> **Answers and Corrections**
>
> Thank you very much for the detailed review of the manuscript.
> We have revisited the manuscript to reflect all the reviewers’ comments. The proposed Representation Dictionary Duality (RDD) concept is explained in detail and the notation inconsistencies throughout the text are corrected. In addition the current literature is updated and the differences between the proposed understanding and the literature is made clear.
> 1) As the reviewer suggests Generative Network (GN) based algorithms do not depend (solely) on PSNR metric. Due to the lack of MSE control, the output is not loyal to the input image. Since textures are created from input images, seemingly randomly, this might cause problems in video streams. Since it is trivial to add Perceptual Loss (PL) minimization to the training procedure, in the future we plan to add PL and conduct experiments.
> 2) We have modified the text to be more comprehensible. We have used the variables g, f and D throughout the text, we have put subscript L for learning (training), and subscript R for reconstruction (testing) phase.
> 3) Similar to 2) we changed section 2.1 to be more comprehensible. We have referred all variables as f.
> 4) The reviewer is correct, not using parenthesis was a typo, thanks for pointing out. It is corrected in the revised text.
> 5) What we meant by “instead of approaching the problem as inverse problem” was to draw attention to the difference of solution approaches of inverse problem solutions and DL based solutions. To avoid misunderstandings we have named the subsections as “Analytic Approaches” and “Data Driven Approaches”. We described dictionary based learning in revised manuscript. Also we added explanations on how Yang et. al. have used LR and HR library for reconstruction.
> 6) The reviewer is correct, this was a typo that we corrected in the revised text.
> 7) We have discussed the effect of size mismatch in the training procedure. Residual learning which we have borrowed from Kim et. al. automatically zero pads the input boundaries and even the outer pixels turn out to be unspoiled. This is added into the text.
> 8) We added a compact image detailing the training of a neural network in appendix.
> 9,10) In Daubechies et. al. from our references, the nature of matrix K is defined as a bounded operator between two hilbert spaces. Boundedness is defined according to the formula: for any given vector f from a Hilbert space, if the inequality ||Kf|| \leq C||f|| is satisfied, where C is a constant, then the operator is bounded. Library D does not violate this assumption, we have added more explanation into the text.
> 11)The RDD concept is tool for explaining how we have incorporated inverse problem and sparse representation mathematics into the CNN training/testing procedure. We have shown that the method, by which the CNN learns, is through solving an inverse problem. The inverse problem constitutes a bounded operator, matrix D, which is composed of LR patches. Even though the matrix D is different in structure from conventional inverse problem operators, it satisfies the constraints to be used as an operator. The cost function that is minimized by CNN training yields a representation vector as the neuron filter, for which the dictionary is matrix D and the target is HR image patch. Neuron parameters (filters) being the representation vectors instead of an output from a network is a new understanding in the literature. Resulting representation vectors (filters) from a layer of neuron filters turn into a dictionary upon which the reconstruction of HR image is carried out during testing phase. This is the core understanding of RDD concept. We moved the explanations given in appendix A into the text.
> 12) For training same 291 images from Kim et. al. have been used in similar fashion, with different rotations and scales. Then we have separated images into two sets by using coherence values from LR patches. We added this information into the text. We will strongly consider jointly optimizing two networks in near future since we already had a goal of finding a better aggregation method.
> 13) For VDSR algorithm Barbara image had 26.2078 dB PSNR and 0.8039 SSIM values whereas our DNSR achieved 26.6600 dB PSNR and 0.8091 SSIM. Cross entropy loss had a minor effect in this improvement.
> 14) Filters might not appear predominant due to the residual learning of the network or because of instanced filters’ size (3x3).
> 15) We have used a foundational paper for mathematical background (Daubechies et. al. 2004) and we have used a state of the art paper covering all previous work including Gregor et. al.’s work (Papyan et. al. 2016,2017). We commented on Gregor et. al.’s paper inside the text and highlight the differences from our approach in revised text. Mainly we show that trained neuron filters become the representation vectors.

---

### Decision · Program_Chairs · 2018-01-29
**ICLR 2018 Conference Acceptance Decision**

**Decision:**

Reject

**Comment:**

This paper addresses the question of how to solve image super-resolution, building on a connection between sparse regularization and neural networks.
Reviewers agreed that this paper needs to be rewritten, taking into account recent work in the area and significantly improving the grammar. The AC thus recommends rejection at this time.